# Can the Necrophagous Blow Fly *Calliphora vicina* (Diptera: Calliphoridae) Be Reared on Plant-Based Meal?

**DOI:** 10.3390/insects15070551

**Published:** 2024-07-21

**Authors:** David F. Cook, Muhammad Shoaib Tufail, Sasha C Voss

**Affiliations:** 1Department of Primary Industries and Regional Development, 3 Baron-Hay Court, South Perth, WA 6161, Australia; shoaib.tufail@dpird.wa.gov.au; 2School of Biological Sciences, The University of Western Australia, 35 Stirling Highway, Crawley, WA 6009, Australia; sasha.voss@uwa.edu.au

**Keywords:** calliphorid, oviparous, larval media, food restriction, larval nutrition, pupation

## Abstract

**Simple Summary:**

The blow fly *Calliphora vicina* has shown potential to be a managed pollination species to support honeybee usage in Australian horticulture. This blow fly species lays eggs onto dead animals (i.e., carrion) shortly after death, and the larvae that hatch feed on the decomposing tissues of the dead animal. However, the use of plant-based media may reduce costs and the unpleasant odor associated with decomposing animal protein when rearing blow flies. Newly hatched larvae of this fly were provided with a diet of either plant-based meals (soya bean and canola) or livestock-based meatmeal; this was done to determine if the plant-based meal could be used to mass-rear this fly for use in horticulture as an insect pollinator of crops. Neither pure soya bean nor pure canola meal diets supported the survival of any larvae through to adult emergence. However, the addition of 10% whole egg powder to the plant-based meals enabled one-quarter of the larvae to survive to adult emergence. Over three-quarters of the larvae survived to adults when reared on a diet of livestock-based meatmeal with the addition of 10% whole egg powder. Larvae fed a diet of livestock-based meatmeal with 10% whole dried egg powder had the fastest development to pupal formation, the highest pupation rate, the heaviest pupae, and the highest subsequent emergence of adult flies, all of which were significantly better than rearing the larvae on a diet with plant-based meals.

**Abstract:**

The use of the blow fly *Calliphora vicina* as a potential pollination species to augment the current reliance on honeybees (*Apis mellifera*) in Australian horticulture requires knowledge of how best to mass-rear this fly species. *Calliphora vicina* lays eggs onto carrion soon after death, and the resultant larvae that hatch are necrophagous and feed on the decomposing tissues of the dead animal. Newly hatched larvae of this fly were provided with plant-based meals (soya bean and canola) and compared with larvae provided with livestock-derived meatmeal to determine if plant-based meal could be used to mass-rear this blow fly species. Both soya bean and canola meal media did not support larval survival through to adult emergence. The addition of only 10% whole egg powder to the plant-based meals enabled survival to eclosion of 39% and 13% on soya bean and canola-based media, respectively, compared with 76% on livestock-based meatmeal with 10% whole egg powder. Larvae fed a diet of livestock-based meatmeal with 10% whole dried egg powder had the fastest development to the pupal stage, the highest pupation rate, the heaviest pupae, and the highest subsequent adult eclosion. This study concluded that the use of plant-based meals as a diet for the mass-rearing of the blow fly *C. vicina* was not a viable option.

## 1. Introduction

Flies are the primary group of insects that are being researched as potential new pollination species [1,2], with the Calliphoridae, Rhiniidae, and Syrphidae families being identified in a review of the role of flies in Australian horticultural crops [3]. There is evidence of pollination by flies in a range of crops, and coupled with this pollination evidence, their foraging behavior, life history traits, and distribution across Australia, 11 calliphorid species were identified as promising candidates as a managed pollination insect in Australia [3]. Two blow fly species have recently been demonstrated as being capable of pollinating avocado trees when placed inside paired-tree enclosures or larger multi-tree enclosures, namely *Calliphora dubia* Macquart 1855 [4] and *Calliphora vicina* Robineau-Desvoidy 1830 [5]. The reliance on honey bees in Australia for pollination needs carries risks associated with the use of a single species. The recent introduction of varroa mite (*Varroa destructor* Anderson & Trueman 2000) into Australia has put further pressure on managed honey bees.

As flies are the second most abundant species to visit flowers [4,6,7,8,9], research has focused on identifying species that could be managed to provide a pollination service, which would involve them being mass-reared to supply as a pollination agent. Some fly taxa are already easily mass-reared with reasonably low inputs and manageable health and safety requirements and present negligible risk of disease transmission to existing wild pollinators when reared under secure colony conditions [3]. The mass-rearing of flies has provided benefits to society across a range of biological and medical fields [10]. The European blue-bottle blow fly (*C. vicina*; formerly known as *C. erythrocephala* [11]) has been identified as a potential insect pollination species for the horticultural industry [3]. Having been introduced into Australia over a hundred years ago, this fly is mostly found in the southern half of Australia.

Under laboratory conditions, *C. vicina* females lay around 500 eggs in their lifetime (over three to four egg masses) [12]; hence, rapidly building up large numbers of this fly is feasible. The rearing of calliphorid larvae has typically used different meat-based products and tissue types primarily from a forensic context to improve the calculation of a post-mortem interval (PMImin) from fly larvae collected on human remains. Variations in the tissue type that blow fly larvae develop on can produce marked differences in the developmental rate and body size, which can compromise predictions of the PMImin within the context of forensic entomology [13,14,15,16]. Most calliphorid larval diets have focused on different animal meats (e.g., pork, beef, lamb, chicken) [17], along with the addition of whole dried egg and milk powder [18,19] or animal fat [16]. However, in an effort to reduce odor some studies have developed artificial diets [20] with varying nutritional profiles [21,22] and the addition of some plant-based sources. For example bran was added to the larval diet of *Lucilia cuprina* Wiedemann 1830 [23] and larvae of *Chrysomya megacephala* Fabricius 1794 were reared on soya flour, milk powder and egg to reduce the odor of putrefied meat [20] and their development was no different to rearing the larvae on a fish meat diet. Green et al. [21] reared black blow fly larvae (*Phormia regina* Meigen 1826) on meridic artificial diets of agar and casein (90% protein and 10% fat) along with cellulose and yeast; the authors noted that cellulose was indigestible to *P. regina*, but did not explain how this finding was determined. Replacement of milk powder with soya bean flour in an artificial diet for rearing screwworm flies (*Cochliomyia hominivorax* Coquerel 1858) resulted in significantly smaller pupae with reduced fitness and fecundity in the emerging adults [24].

Sources of protein fed to laboratory colonies of *C. vicina* for egg development, oviposition, and as a larval-rearing substrate have typically used either pig or cattle tissues (e.g., liver, blood, or muscle) [17,25,26,27,28,29,30]. The rearing of large numbers of calliphorid larvae requires facilities with constant air extraction and ventilation to help reduce the odor of decomposing animal protein sources and ammonia. This study aims to determine if a larval diet of plant-based meals such as soya bean and canola can be used to develop the larvae of a necrophagous blow fly species, in this case, *C. vicina*. There is evidence of some blow fly species being capable of developing from plant material mixed with animal manures [31,32,33] (i.e., reject vegetables fed to cattle) as well as in rare situations from purely rotting vegetable matter associated with vegetable production [32,33].

Soya bean meal and canola meal represent two readily available plant-based livestock feeds that may be able to replace livestock-derived meatmeal in the rearing of a calliphorid fly. Currently, the insect-rearing industry utilizes soya beans as a major source of protein in feeds due to their high content and beneficial composition of amino acids [34]. Protein-rich by-products of the agro-food industry could be used in insect feeds, but it is not known if they also meet the insects’ nutritional requirements [35]. As an example, mealworms have previously been reared on several plant materials, including soya bean meal and canola meal, and their comparable nutritional profile demonstrated the potential to rear them using these three cheap by-products [36,37,38,39]. Both soya bean and canola meal are one of the few vegetable foods that contain all nine essential amino acids [40]. Given the similar nutritional profiles of both soya bean and canola meal, with each containing all the essential amino acids, the hypothesis for the basis of this study was that larvae of a necrophagous fly can be reared on these two plant-based meals to pupation and adult eclosion.

Meatmeal is a product of animal rendering that is often used as a rearing media for calliphorid flies [41,42,43,44] and, more recently, in detailed studies on both *C. vicina* [45] and *C. dubia* [46]. This product is comparatively cheaper than other meat-based media and is available in large quantities with a good blend of protein (50%), carbohydrates (38%), and fat (10%). Adding either whole egg powder or whole eggs (including the shells) to livestock-derived meatmeal larval diets significantly increased the rate of larval development, survival, and adult emergence in both *C. vicina* [45] and *C. dubia* [46]. Whole dried egg powder is in short supply globally, and alternate egg-based sources such as whole eggs discarded by egg layer facilities offer a much cheaper replacement. The costs and logistics involved in mass-rearing each fly species are key factors in deciding what fly species to choose as a managed pollination service. Choosing the most suitable larval-rearing diet involves both the cost of the rearing media and the need to generate high levels of pupation and adult emergence (both >90%). There are multiple examples of rearing *C. vicina* in small-scale laboratory trials using the liver and muscle tissue of various animals. However, these materials are often costly in large volumes, require refrigeration, and produce pungent odors in a rearing facility. For these reasons, livestock-derived meatmeal has often been chosen as the rearing substrate for calliphorid flies.

This study examined the rearing of the oviparous calliphorid *C. vicina* for use as a possible managed pollination species. As the rearing substrate can have a significant effect on larval growth rates [47], this study will determine the nutritional suitability of plant-based meal and livestock-derived meatmeal on the larval development of *C. vicina*. This included measuring the rate of larval growth, the size of the migrating or post-feeding larvae, the number and size (wt) of pupae formed, and the subsequent emergence of adult flies.

## 2. Materials and Methods

### 2.1. Laboratory Colony

A laboratory colony of *C. vicina* was established at the Department of Primary Industries and Regional Development in South Perth, Western Australia. Adult flies were sourced from those caught in the field using carrion-based fly traps with 250 g of beef liver and 125 mL of 1.5% sodium sulfide solution. Solar Fly Traps^®^ company, city, and country, from Arbico Oganics (www.arbico-organics.com, accessed on 19 July 2024) were placed at several locations in the south-west of Western Australia, namely Busselton (−33.64165 S, 115.46172 E), Capel (−33.52121 S, 115.56024 E), and Preston Beach (−32.91854 S, 115.71296 E). Sugar and water were placed within each trap so that live adult flies could survive until being collected several days later. The live flies in the trap were then chilled in a 4 °C cool room so that any adult *C. vicina* could be removed and placed into a separate cage (60 cm × 60 cm × 60 cm). The sex ratio of field-caught adults was not determined as approximately 200 adults were caught and then put into a laboratory at 24.5 °C ± 0.5 °C with 30–40% RH and 14 h light, including a 10 h dark continuous cycle. These flies were then fed beef liver to extract eggs and develop the resultant larvae through to pupation. The eggs extracted from field-caught *C. vicina* were reared on a standard blow fly larval diet of 90% livestock meatmeal and 10% whole dried egg powder (as per *C. dubia* [46] and *C. vicina* [45] in previous studies).

Enough larvae were extracted from the field-caught flies to set up three cages with 500 pupae in each cage (60 cm × 60 cm × 60 cm unit cages), each supplied with water and a 50:50 mixture of sugar and milk powder ad libitum. After adult emergence in each cage, protein was provided twice per week for a period of 24 h as cubes of beef liver sprinkled with blood to enable females to develop eggs.

### 2.2. Larval Extraction

The cages of adult *C. vicina* described above (1 week old and having had two liver feeds at days 2 and 6 after emergence) were presented with beef liver cubes sprinkled with blood on day 9 to elicit oviposition. The liver was checked >26–28 h later to determine the presence of newly hatched, first instar larvae. If present, then 50 larvae were removed using a fine camel-hair paintbrush and placed onto each of five replicates of 200 g of media (i.e., 4 g of media/larvae).

### 2.3. Larval-Rearing Media Composition

The ingredients tested in the present work were from three sources: livestock-derived meatmeal, soya bean meal, and canola meal. The livestock-derived meatmeal was sourced from Talloman Rendering, Hazelmere, WA, Australia, and combined with whole dried egg powder (Farm Pride Foods, Keysborough, VIC, Australia). The Full Fat Soya Bean Meal and Canola Meal were sourced from PBA Feeds, Toowoomba, QLD, Australia. Larvae were reared on a diet of either 100% livestock-derived meatmeal (T1), 90% livestock-derived meatmeal and 10% whole egg powder (T2), 100% soya bean meal (T3), 90% soya bean meal and 10% whole egg powder (T4), 100% canola meal (T5) or 90% canola meal and 10% whole egg powder (T6) in a laboratory-based study. The percentage of each in the respective larval diets is indicated in Table 1, along with the nutritional profile of protein, carbohydrates, and fat. This information was derived from the nutritional analysis provided by the companies that produce each of the products, e.g., Talloman (Hazelmere, WA, Australia) for the livestock-derived meatmeal; PBA Foods (Toowoomba, QLD, Australia) for the soya bean meal and canola meal; and Farm Pride Farms, Keysborough, VIC, Australia) for the whole egg powder. By knowing the proportion of each in the larval diets with two ingredients (i.e., T2, T4, and T6) we could determine the %P, CHO, and F in the diet blend.

Both soya bean and canola meal have a very similar amino acid profile [40] and are both one of the few vegetable foods that contain all nine essential amino acids (Table 2). The levels of each essential amino acid in both soya bean and canola meal protein are close to those of livestock meatmeal [48,49]. Canola meal is a major protein source for animal feeding in Australia because it has high concentrations of protein and a well-balanced amino acid profile [49]. Variation in canola meal protein is a limiting factor in the value of canola meal, where, according to Seberry et al. [50], total crude protein can vary from 36–47%. Protein alone is not a good indicator of canola meal quality, as heat treatment by processes to extract canola oil from seed results in a loss in protein digestibility (see references in [49]).

The dry ingredients of each larval diet were first combined (*v*/*v*) to make up 1 kg of dry media. Water was then added to each dry media blend to make them all up to the same consistency (see Table 1 for amounts added). Five replicate amounts of 200 g of the moist media mixture were then placed into rectangular plastic containers (20 cm × 10 cm wide). Once prepared, 50 newly hatched larvae were then extracted from the liver and placed onto each larval media blend.

The edges of each media tray were cut down to the height of the media so that post-feeding or migrating larvae could easily leave the food source.

### 2.4. Larval Development, Pupation and Adult Emergence

Each larval diet tray was placed onto a 5 cm deep bed of dry sand within a 2 L plastic box with a fine mesh lid secured to capture any adults that emerged. These were kept in a vertical rearing cabinet in the laboratory held at 24.5 °C ± 0.5 °C, 30–40% RH, and 14 h light: 10 h dark continuous cycle. Each larval diet tray was sprayed daily with 25 mL of water to keep the media moist. On noticing that the larval substrates appeared warmer after the first 3 days, I took the temperature in each replicate larval diet tray (n = 30) by placing a thermometer into the center of the media and allowing 30 s for the temperature reading to stabilize. This was recorded for days 3, 4, and 5. Larval migration (i.e., they had left the larval diet and were either under the tray or in the sand), along with newly formed pupae and adult emergence, were recorded every day from every larval diet replicate. This allowed us to examine how larval migration and pupation varied across time for each of the diets tested. The weight (mg) of any migrating larvae and newly formed pupae was recorded from each larval diet. The cohort of weighed migrating larvae and pupae from each day were placed into separate labeled vials within the 2 L plastic box to ensure that the same larvae and/or pupae were only weighed once.

### 2.5. Statistical Analysis

All the data collected on larval development to the migration or wandering phase and pupation was analyzed using R (version 4.1.1) with the “nlme” and “dplyr” packages [52]. An analysis of variance (ANOVA) was performed using the function “aov” [formula: response variable ~ treatment] to evaluate the effects of each treatment (larval media composition; independent variable) on fly development (% larval wanderers, % pupation; dependent variables). After fitting the ANOVA model, the normality and homogeneity of variances assumptions were verified. The normal Q-Q probability plot of residuals was used to check that the residuals were normally distributed, while the residuals versus fits plot was employed to assess the homogeneity of variances. Additionally, Bartlett’s test was used to determine the homogeneity of variances across treatments (i.e., larval diets). Tukey’s HSD Multiple Comparison test was used to determine which means are significantly different from one another at the 5% significance level. In cases where variances were not homogeneous, the Kruskal–Wallis test was employed (non-parametric analysis), and significant differences were identified using Dunn’s test for pairwise comparisons [53]”.

Statistical model selection procedures were conducted to assess the goodness-of-fit to select the best-fitting model. This involved comparing models using the Akaike Information Criterion (AIC), which provides relative information on model fit and allows for model comparison. Lower AIC values indicated better-fitting models. Additionally, dispersion values of less than or equal to 1 were considered in the selection process. Furthermore, model residual diagnostics were performed to evaluate the quality of the model fit. Residuals were examined for patterns or deviations from the assumed model structure. Models with poorly fitting residuals, such as large residuals or systematic patterns, were considered to have a poorer fit and were potentially ruled out [54].

## 3. Results

### Livestock Meatmeal, Soyabean Meal and Canola Meal (T1–T6)

There was a significant difference between the larval diets in development through to post-feeding or migrating larvae (F = 21.139, *df* = 4, *p* < 0.001; Figure 1A). Larval development was most rapid on 90% meatmeal and 10% whole egg powder (T2), where within 7 days peak larval migration of over 80% had occurred. No larvae successfully developed through to the migration stage when fed soya bean meal (T3), and <5% of larvae migrated from a larval diet of canola meal (T5). The addition of 10% whole egg powder to both plant-based meal diets significantly increased the level of larval migration to 29% when added to the canola meal (T6) and 49% when added to the soya bean meal (T4).

The % survival of larvae to pupation was significantly different across larval diets (T1 = 32.8 ± 1.81; T2 = 82.0 ± 4.54; T3 = 0 ± 0; T4 = 46.8 ± 1.91; T5 = 1.6 ± 0.37; T6 = 24.4 ± 5.68; Figure 1B). The variances across the larval diet treatments were not homogenous (Bartlett’s T = 13.64, *df* = 4, *p =* 0.0085), and there was a significant difference between the survival of larvae through to pupation across the larval diets (Kruskal–Wallis, Hc = 23.26, *df* = 5, *p* = 0.0003). Tukey’s HSD procedure indicated that significantly more larvae reared on a diet of meatmeal and egg powder (T2) survived to pupation compared with soya bean (T3) and canola meal (T5; *p* < 0.05). Survival to pupation on a larval diet of soya bean meal and egg powder (T4) was significantly higher than on soya bean meal (T3; Q = 3.05, *p* < 0.005) larval diets T1, T4, and T6 were not significantly different from each other (*p* > 0.05; Figure 1B).

The variances in larval weight (mg) were homogeneous across larval diets (Bartlett’s T = 8.86, *p* = 0.065), and a one-way ANOVA indicated a significant difference between larval diets (F = 72.48, *df =* 4,18, *p* < 0.0001). Mean larval weight was not significantly different (*p* > 0.05) between meatmeal (T1) and meatmeal with whole egg powder (T2) but was significantly different between all other larval diet treatments (*p* < 0.05; Figure 2A). The mean larval weight has been graphed from highest to lowest for ease of interpretation and multiple comparison results. The variances in pupal weight (mg) were also homogeneous across larval diets (Bartlett’s T = 1.86, *p* = 0.761, and a one-way ANOVA indicated a significant difference between larval diet treatments in pupal size (weight in mg; F = 103.05, *p* < 0.0001). Mean pupal weight was significantly different (*p* < 0.05) between each larval diet, with the heaviest pupae in T2, followed by T1, T4, T5, and T6 (Figure 2B). Pupal weight has been graphed from highest to lowest for ease of interpretation and multiple comparison results.

The variances in adult eclosion across the four larval diet treatments where adult eclosion occurred (T1, T2, T4, and T6) were not homogenous (Bartletts T = 19.12, *p* = 0.00026). Hence, a one-way Kruskal–Wallis test indicated that adult eclosion was significantly different across the larval diets (χ^2^ = 14.18, *p* = 0.0027; Figure 3). The lowest adult emergence was from the canola meal and whole egg powder diet (52%, T6), which multiple comparisons indicated was significantly less than both pure livestock meatmeal (90%, T1; Q = 2.889, *p* = 0.004) and meatmeal with 10% whole egg powder (93%, T2; Q = 3.477, *p* = 0.0005; Figure 3).

The adult eclosion rates were graphed from highest to lowest (no adult emergence) for ease of interpretation and multiple comparison results. No adults emerged from either pure soya bean meal (T3) or pure canola meal (T5) larval diets, even though there were 11 pupae formed from T5 (mean wt. 19.3 ± 2.87 mg). The highest adult emergence was from meatmeal and whole egg powder (T2), which was significantly higher (*p* < 0.05) than canola meal and whole egg powder (T6; Figure 3).

An interesting observation during this study was a temporary rise in temperature of the larval-rearing diets and, in particular, the soya bean meal at days 3–5 after placing the larvae onto them (Figure 4) where temperatures were 5–6 °C warmer in the soya bean meal larval diets.

## 4. Discussion

The primary focus of this study was to determine if larvae of the calliphorid blow fly *C. vicina* could be reared on a diet using plant-based meals, as the larval growth phase is usually the most critical phase in the process of mass-rearing flies. Numerous studies in the laboratory have demonstrated that adding whole egg powder to livestock-derived meatmeal resulted in rapid larval migration and pupal formation along with consistently larger pupae and high rates of adult eclosion [45,46]. The addition of only 10% whole egg powder to livestock meatmeal in the larval diet increased pupation by 38%, pupal weight by 16%, and adult emergence by 25%. When rearing the larvae of the calliphorid, *C. dubia* had similar improvements in the same parameters [46]. Several quality control points are measured during mass-rearing, which typically include larval migrant weight, percent pupation and size (weight), and rates of adult eclosion [55]. Other parameters measured post-eclosion include adult flight ability, lifespan, and lifetime fecundity, which were not assessed in this study.

House fly larvae (*Musca domestica* L. 1758) can be reared on plant-based diets consisting of either wheat bran, poultry meal, or soya bean meal, where newly hatched larvae fed soya bean meal had over three-quarters survive to larval migration, of which 95% pupated and all resulted in adult eclosion [56]. By contrast, in this study, no larvae of *C. vicina* survived to the larval migration phase on soya bean meal, and only 4.4% of larvae fed canola meal migrated from the food source and formed very small pupae, with no subsequent adult emergence. One reason for the poor development of *C. vicina* larvae on soya bean meal could be that soya bean has been reported as lacking in the amino acids cysteine and methionine [57] (see Table 1), which are required for animal growth [58]. However, canola meal is reported in the literature as having higher levels of cysteine and methionine than soya bean meal [59] but being limiting in lysine [60]. Methionine and cysteine are the highest sulfur-containing amino acids, whose oxidation results in pungent, volatile sulfur compounds [61]. The fact that no adult flies developed from larvae fed either pure plant meal (soya bean or canola) suggests that some component(s) of animal protein are essential for their development. This was highlighted by the addition of whole egg powder to the plant meals at only 10% of the total larval media, resulting in adult emergence of *C. vicina* (50–80%) from pupae that were formed. This may be a simplistic reason for the lack of development through to adult eclosion when larvae were fed plant-based meals, as [62] showed an interaction between amino acid composition and the microbial population present in larval development of the blow fly *Lucilia sericata* Meigen 1826.

The temporary rise in temperature of the plant-based meal larval diets during the trial (in particular the soya bean meal) did not translate into faster larval development as no larvae survived to pupation on a diet of soya bean meal and <5% survived to pupation on a diet of canola meal. Given that the rate of development of insects, including blow flies is primarily governed by temperature [63] and can differ between even closely related species of blow flies [64], not even this rise in temperature of the media supported faster larval development. Soya bean meal is often fermented to break down the proteins into smaller peptides, which are more easily absorbed by animals and, in the process, generate heat [65].

The black soldier fly, *Hermetia illucens* L. 1758 although capable, like blow flies, of colonizing carrion, is considered a detritivore because its larvae feed on a variety of resources, including plant waste and restaurant food waste, among others [66]. Similar to blow flies, *H. illucens* larvae extend their larval duration to assimilate sufficient nutrients to survive through to pupation and adult emergence [67]. There are marked differences in developmental rate and body size of resultant adults when blow fly larvae are fed different tissue types [17,68,69], which includes *C. vicina*. Larvae of *Ch. megacephala* fed diets high in fat showed increased larval development rates but resulted in smaller adult flies [16]. The type of larval-rearing substrate in this study had a significant effect on the proportion and size of post-feeding larvae, their ability to pupate (along with pupal weight), and successful adult eclosion. Larvae of the calliphorid *Aldrichina grahami* Aldrich 1930 fed pure pork liver paste were significantly heavier than larvae fed a diluted and poorer quality pork liver paste [70]. The odor of decomposing protein when rearing calliphorid flies can be an occupational hazard for workers in a rearing facility. To overcome this, Reddy et al. [20] developed an artificial diet of soya bean flour, milk powder, and whole egg to rear large numbers of *Ch. megacephala* in an effort to reduce the unpleasant odor from the putrefied meat. All life history stages, rate of development, and the resultant size of adults that were produced were no different after rearing this blow fly on the artificial diet as compared to the fish meat diet.

The ability of necrophagous flies to develop from only plant material in natural environments is limited and rarely reported in the literature, where they represented <1% of all fly species that emerged. The only known examples of flies developing from plant-based material include the following: *C. dubia* from rotting snow peas *Pisum sativum* var. *macrocarpum* [32], celery (*Apium graveolens* L.), and cauliflower (*Brassica oleracea* von Plenck) [33]; *Chrysomya rufifacies* Macquart 1842 from rotting cauliflower [32] and leek (*Allium ampeloprasum* L.) [33], and *L. cuprina* from rotting celery [33]. *C. vicina* has previously been reared from rotting residues of beetroot (*Beta vulgaris* L.) [33]. The blow fly species *C. dubia*, *Ch. rufifacies*, and *L. cuprina* are capable of developing in animal manures mixed with plant material. For example, poultry litter (poultry manure and either jarrah (*Eucalyptus marginata* Donn x Smith) or pine (*Pinus radiata* Don. Monterey P.) sawdust) when applied to soil as a fertilizer in vegetable production, [31] where they represented <0.2% of all flies developing from this substrate. *C. dubia* adults developed from reject vegetables when fed to livestock and mixed with animal manure [32], in particular, reject cauliflowers fed to cattle (17.3%), but also carrots fed to cattle (<0.01%).

Mass-rearing of house flies is possible on plant-based substrates consisting of 50% wheat bran, 30% alfalfa meal, and 20% corn meal, which produced the highest survival to pupation and heaviest pupal weights compared with animal wastes (e.g., dairy, swine, or poultry manure) [71]. Pérez et al. [12] reared larvae of *C. vicina* on several different artificial diets; the shortest developmental time (egg to pupae) was on pig’s liver (18.8 days) compared with milk powder and egg (24.6 days) and powdered liver. Our studies showed *C. vicina* development from egg to pupae took only 9–10 days on a larval diet of meatmeal and whole egg powder, which is less than half the duration recorded by Pérez et al. [12], who reared *C. vicina* larvae at 26 ± 2 °C, 70 ± 10% RH and 12:12 h (L:D). *Calliphora vicina* has been described as highly adaptable to being reared in a laboratory setting with artificial nutritional larval diets [12]. The study reported here only partly supports this description, where larval diets consisting of 90% meatmeal and 10% whole egg powder ensured rapid larval development to pupation, the highest pupation rate, and subsequent adult emergence. When reared on a diet of plant-based meal and whole egg powder, both larval development and survival of pupae to adult eclosion were significantly impaired.

This study demonstrated that no *C. vicina* larvae were capable of developing through to pupation when reared on a larval diet of soya bean meal. Although there was some level of pupal formation on canola meal (<5%), no adults emerged from these pupae. The addition of just 10% whole egg powder to the plant-based meal in their larval diets enabled between 30–50% of larvae to migrate from the media and 20–40% to then pupate with some adult emergence from the soya bean meal and 10% whole egg powder (T4) (39% of initial larvae) and less so from canola meal and 10% whole egg powder (T6) (13% of initial larvae). Hence, the use of plant-based meals as ingredients to rear *C. vicina* larvae in any mass-rearing scenario has not been supported by this study, where neither pure plant meals supported larval development through to adult eclosion. Even when blended with 10% whole egg powder, the plant meal media resulted in lighter-weight larvae (26% soya bean meal and 50% canola meal), lighter-weight pupae (22% soya bean meal and 52% canola meal), and reduced adult emergence (51% soya bean meal and 83% canola meal) when compared with livestock meatmeal and whole egg powder.

## Figures and Tables

**Figure 1 insects-15-00551-f001:**
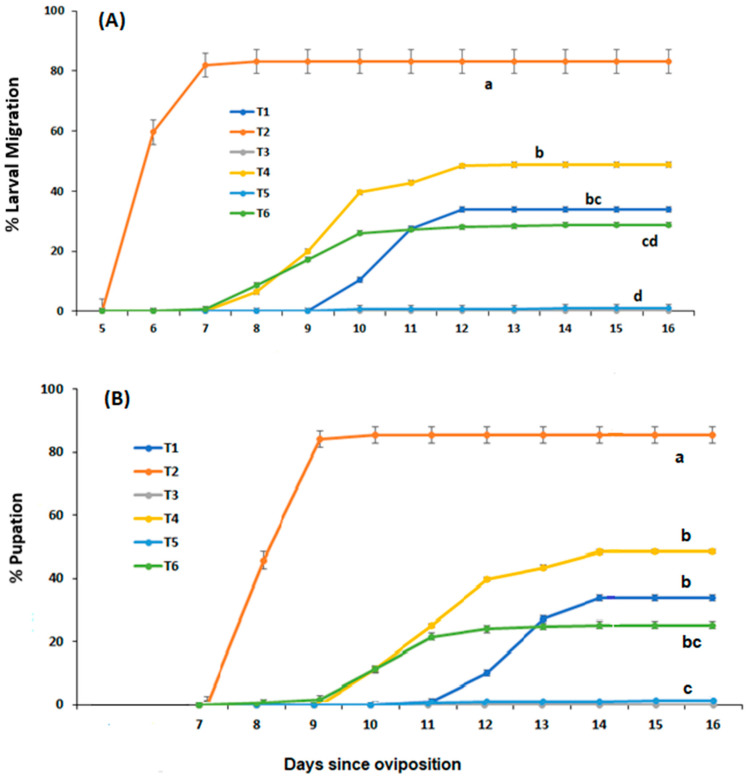
Proportion of *C. vicina* larvae that had (**A**) migrated off the dietary media and (**B**) pupated up to 16 days since oviposition when reared on either: T1 = 100% meatmeal; T2 = 90% meatmeal and 10% whole egg powder; T3 = 100% soya bean meal; T4—90% soya bean meal and 10% whole egg powder; T5 = 100% canola meal; or T6 = 90% canola meal and 10% whole egg powder. The bars indicate the standard error, and the different letters indicate significant differences between larval diets (*p* ≤ 0.05, Tukey HSD test).

**Figure 2 insects-15-00551-f002:**
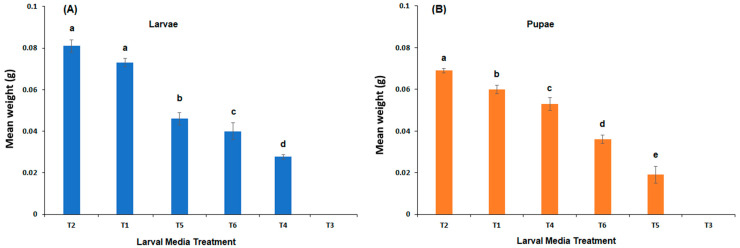
Mean weight ± s.e. of *C. vicina* migrating larvae (**A**) and pupae (**B**) from newly hatched larvae fed a diet of either: 100% meatmeal (T1); 90% meatmeal and 10% whole egg powder (T2); 100% soya bean meal (T3); 90% soya bean meal and 10% whole egg powder (T4); 100% canola meal (T5); or 90% canola meal and 10% whole egg powder (T6). The bars indicate the standard error and the different letters indicate significant differences between larval diets (*p* ≤ 0.05, Tukey HSD test).

**Figure 3 insects-15-00551-f003:**
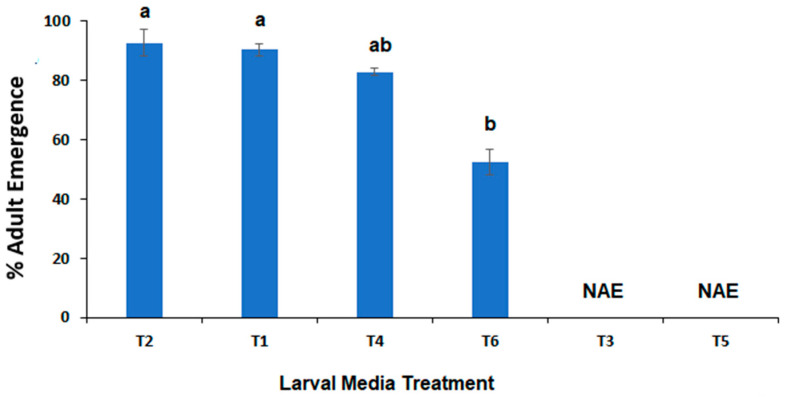
Mean % adult emergence of *C. vicina* from newly hatched larvae fed a diet of either: 100% meatmeal (T1); 90% meatmeal and 10% whole egg powder (T2); 100% soya bean meal (T3); 90% soya bean meal and 10% whole egg powder (T4); 100% canola meal (T5); or 90% canola meal and 10% whole egg powder (T6). The bars indicate the standard error and the different letters indicate significant differences between treatments (*p* ≤ 0.05, Tukey HSD test). NAE = No adult emergence.

**Figure 4 insects-15-00551-f004:**
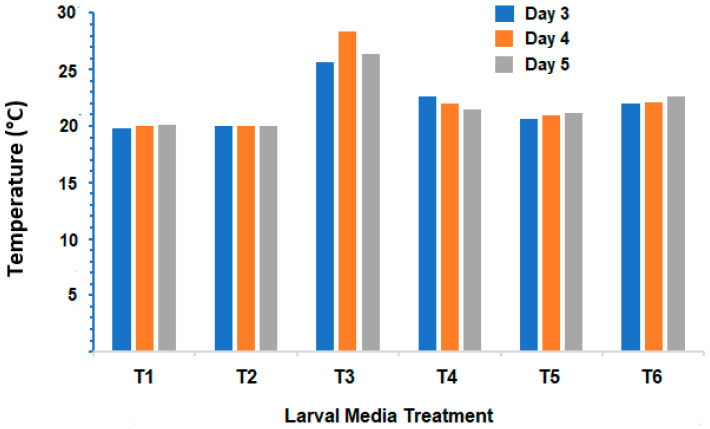
Mean temperature of the larval-rearing media at days 3–5 after placing newly hatched larvae of *C. vicina* onto a dietary blend of either: T1 = 100% meatmeal; T2 = 90% meatmeal and 10% whole egg powder; T3 = 100% soya bean meal; T4 = 90% soya bean meal and 10% whole egg powder; T5 = 100% canola meal; or T6 = 90% canola meal and 10% whole egg powder.

**Table 1 insects-15-00551-t001:** Proportion of each ingredient (*v*/*v*) (dry) in a range of larval diets (T1–T14) tested across three laboratory-based studies using newly hatched larvae of *Calliphora vicina*. The proportions of protein (P), carbohydrates (CHO), and fat (F) in each diet are indicated.

	% Larval Diet Ingredients (Dry)				Water	Nutritional Profile		
Treatment	LMM	WEP	SBM	CM		%P	%CHO	%F
T1	100				200 mL	52.0	38.0	10.0
T2	90	10			200 mL	51.4	34.5	13.2
T3			100		600 mL	47.5	32.5	18.0
T4		10	90		600 mL	47.4	29.6	20.4
T5				100	800 mL	36.9	33.6	12.0
T6		10		90	800 mL	35.2	30.3	15.0

LMM = livestock meatmeal; WEP = whole egg powder; SBM = soya bean meal; CM = canola meal; P = protein; CHO = carbohydrates; F = fat.

**Table 2 insects-15-00551-t002:** Amino acid profile of livestock-derived meatmeal and both soya bean and canola meal *.

	Livestock Meatmeal	Soya Bean Meal	Canola Meal
Essential amino acids			
Arginine	4.80	7.20	5.80
Histidine	1.44	2.60	2.70
Iso-leucine	1.87	4.00	4.00
Leucine	4.16	7.80	7.00
Lysine	3.64	6.40	5.80
Methionine	1.11	1.30	1.90
Phenylalanine	2.29	5.00	3.80
Threonine	2.31	4.00	4.50
Valine	2.69	4.80	5.00
Non-essential amino acids			
Alanine	5.16	4.30	4.30
Aspartic acid	5.18	11.70	7.00
Cystine	1.03	0.64	
Glutamic acid	8.83	18.70	17.50
Glycine	9.33	4.20	4.90
Proline	6.04	5.10	6.00
Serine	2.66	5.10	4.60
Tyrosine	1.57	3.20	3.10

* Livestock meatmeal data was sourced from [51], and plant-based meal data was sourced from [40]. Values are expressed as % of crude protein.

## Data Availability

The data that support this study are openly available from the corresponding authors (DC, ST, and SV) and stored at the University of Western Australia Data Repository under “Rearing blowflies (Diptera: Calliphoridae) on plant-based meal” The University of Western Australia, 2024 (DOI: 10.26182/3vtq-a713).

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
