# Peer review of "Can the Necrophagous Blow Fly Calliphora vicina (Diptera: Calliphoridae) Be Reared on Plant-Based Meal?"

_insects, 2024, doi:10.3390/insects15070551_

Round 1

Reviewer 1 Report

Comments and Suggestions for Authors

Excellent manuscript and well written. I have no comments or suggestions for improvement other than a couple of editorial notes.

Figures 1-3 should indicate in the legend or figure itself what the bars on the columns mean

Is it 'soyameal' or 'soya meal'  Inconsistent use.

Otherwise, very well done

Author Response

Figures 1-3 should indicate in the legend or figure itself what the bars on the columns mean

Response: I have indicated in each of Figures 1, 2 and 3 that the bars indicate the standard error.

Is it 'soyameal' or 'soya meal'  Inconsistent use.

Response: I have gone through and consistently used the term “soya bean meal” throughout the paper.

Reviewer 2 Report

Comments and Suggestions for Authors

The authors are to be complimented on a solid research project. The experiments are straight forward and explained clearly.  The results are solid and the knowledge generated from this study will assist in mass rearing of Calliphora vicina. There are a few issues with the structure of the paper that need to be addressed.  The introduction needs to be laid out in a more logical order - I have put further comments in the PDF.  There are results that are discussed for the first time in the Discussion section and they should be moved to the Results section - I have indicated this in the PDF.  

Comments on the Quality of English Language

The English used is acceptable.  There are a few minor corrections indicated in the PDF.

Author Response

There are a few issues with the structure of the paper that need to be addressed.  The introduction needs to be laid out in a more logical order - I have put further comments in the PDF. 

Response: I removed the first sentence of the last paragraph “As opposed to house fly larvae (Musca domestica L. 1758) which can be reared on plant-based diets consisting of either wheat bran, poultry meal or soya bean meal [40], blow fly larvae have typically been reared on animal tissues and associated products (e.g., blood, milk powder, eggs (whole dried powder or whole eggs)).” 

The last paragraph of the introduction now reads

This study examined rearing of the oviparous calliphorid C. vicina for use as a potential managed pollination species. As the rearing substrate can have a significant effect on larval growth rates [41] this study will determine the nutritional suitability of plant-based meal and livestock meatmeal on larval development of C. vicina. This included measuring the rate of larval growth, the size of the migrating or post-feeding larvae, the number and size (weight) of pupae formed, and the subsequent emergence of adult flies.

Indicate what temperature regime you kept the adult fly colony at along with the larval rearing conditions

Response: In the M&M Section 2.1 Laboratory Colony, I have indicated the mean temperature and range, the relative humidity and photoperiod at which adult flies were kept in the laboratory, viz, "the field adults caught were kept in the laboratory at 24.5 °C ± 0.5 °C, 30–40% RH and 14 h light: 10 h dark continuous cycle"

Section 2.4 Larval Development, Pupation and Adult Emergence

Response: I clarified in the opening sentence that each tray of larval media was placed onto a 5 cm deep bed of dry sand within a 2 L plastic box “with a fine mesh lid secured to capture any adults that emerged”

How did I make sure that I didn’t weight the same larvae/pupae more than once?

Response: I clarified in the methodology by adding the statement: “Each day all migrating larvae and pupae that were recorded and weighed were placed into a vial with sand within the 2L plastic box to ensure they weren’t weighed more than once”

There are results that are discussed for the first time in the Discussion section and they should be moved to the Results section - I have indicated this in the PDF.  

Response: I removed the paragraph on page 7 line 11 and pasted it into the M&M section as suggested by the reviewer as it discusses the nutrient profile of each of the soya bean and canola meals.  

In addition, I also moved the finding about the elevated temperatures in the soya bean meal media for several days to the end of the results sections as suggested by the reviewer, which included Figure 4.

What temperature did Pérez et al. (2016) rear Calliphora vicina larvae at?

Response: I clarified in the discussion that Pérez et al. (2016) reared C. vicina larvae at 26 ± 2° C, 70 ± 10 % relative humidity, and a photoperiod of 12:12 (L:D) h .

I have attached all responses to those marked in the pdf of the MS in the file uploaded against this reviewer's comments.  

Reviewer 3 Report

Comments and Suggestions for Authors

The authors present a novel and unique approach to pollination, using a species commonly solely used for forensic analysis. This approach bears scientific validity, even though the results show that it is not a viable option for pollination. 

Please see detailed comments below:

Page 2 line 15: No need to reference C. erythrocephala – this notation has not been used in literature for many years; C. vicina is common

Page 2 line 17: change “most” to “mostly”

Page 2 line 22: Use correct terminology (i.e. PMImin) to denote post mortem interval

Page 3 line 28: Add the website/URL in as a reference and cite in text, rather than add URL in-text

Page 3 line 31: Change “till” to “until”

Page 3 section 2.1: No mention of sexing flies? Were collected C. vicina flies sexed or all placed into a rearing container and hoping that there was a decent male:female ratio present for reproduction purposes? Please clarify.

Page 3 line 38. The authors mention that adult C. vicina were approximately 1 week old – how was this determined? Does this refer to 1 week in captivity, as there is no way to determine the age of the adults when they were trapped in the field? Please clarify.

 The results and discussion are clearly presented and conclusions support these. I would advise perhaps expanding the discussion to include a section on black soldier fly, as this species is also capable of feeding on carrion and plant-derived diets. but this is not critical to the validity of the paper. 

Comments on the Quality of English Language

Minor grammatical edits are required but overall the quality of English is acceptable and understandable. 

Author Response

Page 2 line 15: No need to reference C. erythrocephala – this notation has not been used in literature for many years; C. vicina is common

Response: I have left the reference in for C. vicina being formerly known as C. erythrocephala, which I believe is important for readers unfamiliar with this fly species to look up older references. 

Page 2 line 17: change “most” to “mostly”

Response: I have changed to “mostly” in the MS

Page 2 line 22: Use correct terminology (i.e. PMImin) to denote post mortem interval

Response: I have changed to “PMImin” in the MS

Page 3 line 28: Add the website/URL in as a reference and cite in text, rather than add URL in-text

Response: I have left the website address in the text rather than re-numbering all the references – does not make much difference at all. It is standard in the methodology of many papers that materials sought for the study have the company website address. 

Page 3 line 31: Change “till” to “until”

Response: I have changed to “until” in the MS

Page 3 section 2.1 Laboratory Colony. No mention of sexing flies? Were collected C. vicina flies sexed or all placed into a rearing container and hoping that there was a decent male:female ratio present for reproduction purposes? Please clarify.

Response: There was no sexing of adult flies as several hundred individuals were caught from the field and placed into a cage to extract eggs for setting up a larger adult colony to use for this trial. This was clarified in the methodology section of 2.1.

Page 3 line 38. The authors mention that adult C. vicina were approximately 1 week old – how was this determined? Does this refer to 1 week in captivity, as there is no way to determine the age of the adults when they were trapped in the field? Please clarify.

Response: I have clarified in the M&M that the first collection of field adults were fed liver to extract eggs and produce thousand pupae for distribution amongst unit cages of 500 adults.  These cages of adult flies had their emergence date recorded and were then protein fed twice over approx. 1 week for extraction of newly hatched larvae for use in the experiments described.

The results and discussion are clearly presented and conclusions support these. I would advise perhaps expanding the discussion to include a section on black soldier fly, as this species is also capable of feeding on carrion and plant-derived diets. but this is not critical to the validity of the paper. 

Response: I added a small section on the black soldier fly in the discussion, viz,

The black soldier fly, Hermetia illucens L. 1758 although capable, like blow flies, of colonizing carrion, is considered a detritivore as it’s larvae feed on a variety of resources including plant waste and restaurant food waste amongst others [66]., Similar to blowflies, H. illucens larvae extend their larval duration to assimilate sufficient nutrients to survive through to pupation and adult emergence [67].  

The formatting of the introduction is not logically laid out. Start with stating the problem, provide the supporting information and then move onto discussing C. vicina and why you did the research.  All the information is there it is just not structured in the best way.

Response: The introduction has been altered previously as per Reviewer’s 1-3 comments and better sets out a logical order

The start of the introduction implies that flies are leading the research, please rephrase

Response: I have changed the start of the introduction to now state “Flies are the primary group of insects that are being researched as potential new pollination species”

Reviewer 4 Report

Comments and Suggestions for Authors

Review of manuscript insects-3025044

The manuscript by Cook and colleagues reports results of a study that tested plant-based (soybean and canola meals) and animal-based (meatmeal) diets for rearing the necrophagous calliphorid fly Calliphora vicina, a potential pollinator of crop plants in Australia.  The topic of the manuscript is relevant and interesting in insect science.  The use of plant-based diets could have important practical applications for mass rearing of C. vicina.  I feel that the topic of this manuscript fits nicely the remit of the topical collection "Science of Insect Rearing Systems", and that many readers would appreciate one such a paper included in this collection.

Despite the overall merit of the study, I found that the manuscript has several weaknesses requiring the attention of the authors before publication.  Below I provide general and specific comments that I feel can help authors to improve their manuscript.

General comments:

The Introduction presents relevant information about the study system, but it fails to present readers the functional idea that the authors tested with their research.  That is, what was the hypothesis and predictions the authors tested in their study.  Unfortunately for science, it is has become common practice for researchers and journals to publish experimental studies lacking hypothesis and predictions, thereby, leaving aside the hypothetical-deductive method that should govern scientific thinking.  In addition, at the end of the introduction (p. 3 lines 17-19) it is mentioned "this study will determine the nutritional suitability of plant-based meal and livestock meatmeal on larval development of C. vicina", but the nutritional content of the diets were not manipulated and there is no clear explanation about what nutrients were responsible of making one or another diet suitable for rearing this species.

My major concern with this manuscript relates to a vague and imprecise description of the experimental design and protocol.  There is no section devoted to clearly explain the experimental design, including description of the independent and dependent variables with their levels and measurement scale, the experimental units, the response unit, the unit of replication, randomization, the distribution of experimental units in time and space, the number of replicates, grouping structure (if any), etc.  Some of the information is somehow present in the manuscript, but the way it is presented does not allow an easy understanding on how the experiment was designed and conducted.  In section "2.3. Larval Rearing Media Composition", it is mentioned that "The mixture was then divided across each replicate...", but it is not mentioned in the manuscript how many times each diet treatment was replicated.  This information needs to be included in the manuscript.

The rationale for choosing a specific statistical test should be based on the nature and scale of the independent and dependent variables, and the objectives of the research.  But the lack of a clear explanation of the experimental design in this manuscript, limits appreciation of the statistical analyses and the results.

The section "2.5. Statistical Analysis" needs several improvements: (i) you need to indicate what specific statistical model was fitted with "the “nlme” package" and why this specific model (and library) was appropriate for the data on larval development to the wandering phase and pupation.  (ii) Please be aware that checking "normality of raw data in general linear models" as is anova, is "the most widespread myth in statistics" (Kéry, M., & Hatfield, J. S. 2003. Normality of raw data in general linear models: the most widespread myth in statistics. Bulletin of the Ecological Society of America, 84, 92-94); you need to correct this mistake.  In page 4, lines 23-24, it is mentioned that ANOVA was used "If the data was normally distributed", but basing the choice of using anova on the normality of the data is not statistically valid.  As a reminder to the authors, ANOVA is used when the independent variable is categoric with three or more levels, and the dependent variable is continuous numerical.  After fitting the model to the data, the normal distribution and equal variance of the residuals (NOT of the data) should be examined.  These are assumptions that must be met when fitting linear models.  If, after fitting ANOVA, you found a violation of the normality and/or equal variance of the residuals assumptions, then you can choose to transform the dependent variable or to use a non-parametric test.  However, please note that non-parametric tests do not compare means-as the name indicates, non-parametric tests does not estimate parameters such as means.  As such, the statement "non-parametric Kruskal-Wallis test was used to compare group means" is misleading.

So many gaps in the description of the methods, especially in the description of the experimental design and statistical analyses, limits understanding of the results.  For instance, in Figure 1, the variable "days since oviposition" is presented as a covariate.  But before it is not clearly stated that there was an interest in examining how larval migration and pupation varied across time for each of the diets tested.  Furthermore, the description of statistics in the results section is incomplete.  For instance, in p. 4 line 34, the anova statistics should include degrees of freedom for both the treatment and the replicates.  In p. 5, lines 9 and 14, there are no degrees of freedom reported for ANOVA.

Bar plots in Figures 2 and 3 show the diet treatments on the y-axis ordered from the highest to the lowest values of the response variable.  But it is not explained before in the text or in the figure legends why this order was followed instead of the logic order of T1-T6.  I think it would be better to present the treatments ordered from T1 to T6.

The discussion starts by saying that "This primary focus of this study was to optimize the rearing of the calliphorid blow fly C. vicina...".  Regardless of the unclear wording, it is not true that the focus of the study was to optimize the rearing of C. vicina.  There is no mention in the introduction or the materials and methos sections that the goal of the study was to optimize the rearing or how optimization was performed.  Optimization relates to an advanced step in a sequential process of experimentation that was not followed in this research.

The discussion is not the place to present results (i.e., Table 1 and figure 4).

Overall, I feel this manuscript fails to flush out the significance of the study and that authors need to be more precise and specific with their writing.

A final recommendation to authors is to use consecutive line numbering when submitting a paper for peer review.  This manuscript restarts line numbering in each page, adding unnecessary complexity to the work of reviewers.

Specific comments:

Page 1, line 38: you cannot conclude that plant-based meals were not economically viable as the manuscript does not present an economic analysis.

Page 2, lines 33-34: the definition of meridic diet is not a diet "not containing any insect components".  Please check Cohen, A. C. (2003). Insect diets: science and technology. CRC press, for a definition of meridic diet.

In section "2.1. Laboratory Colony" (p. 3), please explain how the larvae were reared.  What larval diet was used for the maintenance of the colony?

Page 4, lines 5-6: "sufficient water" is vague and imprecise.  Indicate, exactly, how much water was added to the dry ingredients.  Include a Tabel with diet formulations as percentages.  You should include water in the diet formulations.

Page 9, lines 7-8, it is mentioned that "The ability of necrophagous flies to develop from only plant material is limited and rarely reported in the literature".  So, why did you tested plant-based diets in the first place.  Need to strength the rationale for testing soy bean and canola meals.

Author Response

The Introduction fails to present readers the functional idea that the authors tested with their research.  That is, what was the hypothesis and predictions the authors tested in their  study.  Unfortunately for science, it is has become common practice for researchers and journals to publish experimental studies lacking hypothesis and predictions, thereby, leaving aside the hypothetical-deductive method that should govern scientific thinking.  

Response: This would require a major overhaul of the paper to satisfy what is an historic and in a sense quite out-dated approach to scientific experimentation. The hypothesis to test was clear – can C. vicina larvae be reared through to pupation and adult emergence on plant based meal, specifically soya bean meal and canola meal.

There needs to be more explanation of the BASIS of the selected plant substitutes for meat products. By “basis” I mean a statement of rationale as to how and why the selected plant substitutes were chosen for experimentation. This would involve some kind of statement about the nutritional value of the various components, including some kind of proximate analysis/composition of the products and the diet mixtures. In conjunction with the rationale based on composition/proximate analysis, the paper would profit from a statement of hypothesis as to which combinations could be expected to yield the best responses to the component manipulations.

Response: I have inserted a new paragraph in the Introduction on page 2, starting line 52.  This explains why I chose to test both soya bean meal and canola meal in this study, which reads:

Soya bean meal and canola meal represent two readily available plant based livestock feeds that may be able to replace animal-derived meatmeal in the rearing of a calliphorid fly. Currently, the insect rearing industry utilizes soya bean as a major source of protein in the feeds due to their high content and beneficial composition of amino acids [34]. Protein-rich by-products of the agro-food industry could be used in insect feeds, but it is not known if they also meet the insects’ nutritional requirements [35]. As an example, mealworms have previously been reared on several plant materials including soya bean meal and canola meal where their comparable nutritional profile demonstrated the potential to rear mealworms using these three cheap by-products [36-39]. Both soya bean and canola meal are one of the few vegetable foods that contain all 9 essential amino acids [40].

There is already a profile of the nutritional components of both canola and soya bean meal in Table 1 in the Materials and Methods section 2.3.

In addition, at the end of the introduction (p. 3 lines 17-19) it is mentioned "this study will determine the nutritional suitability of plant-based meal and livestock meatmeal on larval development of C. vicina", but the nutritional content of the diets were not manipulated and there is no clear explanation about what nutrients were responsible of making one or another diet suitable for rearing this species.

Response: No the nutritional contents of the larval diets were not manipulated by adding or subtracting various nutritional macro nutrients, but each media blend was described in terms of nutrition available to the feeding larvae based on the nutritional profile of each media ingredient. 

My major concern with this manuscript relates to a vague and imprecise description of the experimental design and protocol.  There is no section devoted to clearly explain the experimental design, including description of the independent and dependent variables with their levels and measurement scale, the experimental units, the response unit, the unit of replication, randomization, the distribution of experimental units in time and space, the number of replicates, grouping structure (if any), etc.  Some of the information is somehow present in the manuscript, but the way it is presented does not allow an easy understanding on how the experiment was designed and conducted.  In section "2.3. Larval Rearing Media Composition", it is mentioned that "The mixture was then divided across each replicate...", but it is not mentioned in the manuscript how many times each diet treatment was replicated.  This information needs to be included in the manuscript.

Response:

The other 3 reviewers all had no issue with the materials and method section of this paper. 

Reviewer 1 stated “Excellent manuscript and well written. I have no comments or suggestions for improvement”

Reviewer 2 stated “The authors are to be complimented on a solid research project. The experiments are straight forward and explained clearly”

Reviewer 3 stated “The authors present a novel and unique approach to pollination, using a species commonly solely used for forensic analysis. This approach bears scientific validity..”

It is clearly mentioned twice in the M&M section that 5 replicates of 200g of each media blend were set up (each with 50 newly hatched C. vicina larvae on them)

I have made considerable changes to the M&M section based on the other 3 reviewer’s comments so it is a lot clearer

The rationale for choosing a specific statistical test should be based on the nature and scale of the independent and dependent variables, and the objectives of the research.  But the lack of a clear explanation of the experimental design in this manuscript, limits appreciation of the statistical analyses and the results.

Response: The experimental design in this study is cleared explained as evidenced by no other reviewer having a problem with the experimental design.

The section "2.5. Statistical Analysis" needs several improvements: (i) you need to indicate what specific statistical model was fitted with "the “nlme” package" and why this specific model (and library) was appropriate for the data on larval development to the wandering phase and pupation.  (ii) Please be aware that checking "normality of raw data in general linear models" as is anova, is "the most widespread myth in statistics" (Kéry, M., & Hatfield, J. S. 2003. Normality of raw data in general linear models: the most widespread myth in statistics. Bulletin of the Ecological Society of America, 84, 92-94); you need to correct this mistake.  In page 4, lines 23-24, it is mentioned that ANOVA was used "If the data was normally distributed", but basing the choice of using anova on the normality of the data is not statistically valid.  As a reminder to the authors, ANOVA is used when the independent variable is categoric with three or more levels, and the dependent variable is continuous numerical.  After fitting the model to the data, the normal distribution and equal variance of the residuals (NOT of the data) should be examined.  These are assumptions that must be met when fitting linear models.  If, after fitting ANOVA, you found a violation of the normality and/or equal variance of the residuals assumptions, then you can choose to transform the dependent variable or to use a non-parametric test.  However, please note that non-parametric tests do not compare means-as the name indicates, non-parametric tests does not estimate parameters such as means.  As such, the statement "non-parametric Kruskal-Wallis test was used to compare group means" is misleading.

Response: The statistical analysis section including the choice of R version, the packages used and the reason for performing ANOVA on normally distributed data along with defining the independent and dependent variables.  The tests used for variance homogeneity and multiple comparison tests are fully explained.  An extra paragraph has been added on the statistical model selection procedures used for the analyses of the data in this paper.  So the section 2.5 Statistical Analysis now reads:

All the data collected on larval development to the wandering phase and pupation was analysed using R (version 4.1.1) with the “nlme” and “dplyr” packages [46]. An analysis of variance (ANOVA) was performed for normally distributed data using the function “aov” [formula: response variable ~ treatment] to evaluate the effects of each treatment (larval media composition; independent variable) on fly development (% larval wanderers, % pupation; dependent variables). Bartlett’s test was used to determine the homogeneity of variances across treatments and then Tukey’s Multiple Comparison test to determine which means are significantly different from one another at the 5% significance level. In cases where variances were not homogenous, the non-parametric Kruskal-Wallis test was employed to compare group means. Subsequently a post-hoc test was conducted for pairwise comparisons to identify significantly different means [47].

  Statistical model selection procedures were conducted to assess the goodness-of-fit to select the best-fitting model. This involved comparing models using the Akaike Information Criterion (AIC), which provides relative information on model fit and allows for model comparison. Lower AIC values indicated better-fitting models. Additionally, dispersion values of less than or equal to 1 were considered in the selection process. Furthermore, model residual diagnostics were performed to evaluate the quality of the model fit. Residuals were examined for patterns or deviations from the assumed model structure. Models with poorly fitting residuals such as large residuals or systematic patterns were considered to have poorer fit and were potentially ruled out [48].

Comment: So many gaps in the description of the methods, especially in the description of the experimental design and statistical analyses, limits understanding of the results.  For instance, in Figure 1, the variable "days since oviposition" is presented as a covariate.  But before it is not clearly stated that there was an interest in examining how larval migration and pupation varied across time for each of the diets tested.  Furthermore, the description of statistics in the results section is incomplete.  For instance, in p. 4 line 34, the anova statistics should include degrees of freedom for both the treatment and the replicates.  In p. 5, lines 9 and 14, there are no degrees of freedom reported for ANOVA.

Response: I have indicated the df (5) in the text for the Kruskal-Wallis non parametric ANOVA on larval survival to pupation and

Response: I have Indicated the df (4, 18) in the text for the ANOVA on larval weight.

Bar plots in Figures 2 and 3 show the diet treatments on the y-axis ordered from the highest to the lowest values of the response variable.  But it is not explained before in the text or in the figure legends why this order was followed instead of the logic order of T1-T6.  I think it would be better to present the treatments ordered from T1 to T6.

Response: I have clarified when describing the results depicted in Figures 2 and 3 that the mean for each parameter (i.e., larval weight, pupal weight and % adult eclosion) that the data has been graphed from highest to lowest for ease of interpretation and multiple comparison results.

The discussion starts by saying that "This primary focus of this study was to optimize the rearing of the calliphorid blow fly C. vicina...".  Regardless of the unclear wording, it is not true that the focus of the study was to optimize the rearing of C. vicina.  There is no mention in the introduction or the materials and methods sections that the goal of the study was to optimize the rearing or how optimization was performed.  Optimization relates to an advanced step in a sequential process of experimentation that was not followed in this research.

Response: I agree with the reviewer’s comment and have changed the opening sentence of the discussion to now read: “The primary focus of this study was to determine if larvae of the calliphorid blow fly C. vicina could be reared on plant-based meals, as the larval growth phase is usually the most critical phase in the process of mass rearing flies.”

The discussion is not the place to present results (i.e., Table 1 and figure 4).

Response: I have dealt with this issue previously as per Reviewer 2 comments, where I moved the finding about the elevated temperatures in the soya bean meal media for several days to the end of the results sections as suggested by the reviewer, which included Figure 4.

Overall, I feel this manuscript fails to flush out the significance of the study and that authors need to be more precise and specific with their writing.

Response: I consider the significance of this study to be clearly articulated in the discussion. 

A final recommendation to authors is to use consecutive line numbering when submitting a paper for peer review.  This manuscript restarts line numbering in each page, adding unnecessary complexity to the work of reviewers.

Response: The numbering of each page was set out by the Journal with line numbering starting at 1 for each page.

Specific comments:

Page 1, line 38: you cannot conclude that plant-based meals were not economically viable as the manuscript does not present an economic analysis.

Response: I changed the last sentence of the abstract to now state: This study concluded that the use of plant-based meals in the mass rearing of the blow fly C. vicina was not a viable option.

Page 2, lines 33-34: the definition of meridic diet is not a diet "not containing any insect components".  Please check Cohen, A. C. (2003). Insect diets: science and technology. CRC press, for a definition of meridic diet.

Response: I removed the definition of what a meridic diet was as the paper by Green et al (2003) on rearing of the black blowfly using artificial diets is adequate

In section "2.1. Laboratory Colony" (p. 3), please explain how the larvae were reared.  What larval diet was used for the maintenance of the colony?

Response: I have explained that the first cohort of field caught C. vicina were fed beef liver and blood to extract eggs, which were then placed onto a media blend of 90% livestock derived meatmeal and 10% whole egg powder, which in two recent previous studies was shown to be optimal for larval development, pupation and adult eclosion of both C. vicina and C. dubia, which is the standard diet for “maintenance of the colony”. 

Page 4, lines 5-6: "sufficient water" is vague and imprecise.  Indicate, exactly, how much water was added to the dry ingredients.  Include a Tabel with diet formulations as percentages.  You should include water in the diet formulations.

Response: I have inserted in the last paragraph of Section 2.4 Larval Media Composition the amount of water added to each media treatment to make up media blends with a similar consistency i.e. T1 and T2 had 200mL of water added; T3 and T4 had 600mL of water added; T5 and T6 had 800mL of water added. 

Page 9, lines 7-8, it is mentioned that "The ability of necrophagous flies to develop from only plant material is limited and rarely reported in the literature".  So, why did you tested plant-based diets in the first place.  Need to strength the rationale for testing soy bean and canola meals.

Response: I clarified that the ability of necrophagous flies to develop from only plant material in natural environments is limited and rarely reported in the literature. I mention several times in the paper that plant-based meals were tested as a method for the mass rearing of a necrophagous calliphorid, fly, to reduce the offensive odours of decomposing animal proteins and to find local, reliable and consistent ingredients for integration into their mass rearing. 

Round 2

Reviewer 4 Report

Comments and Suggestions for Authors

Asking authors to include explicit mention of the functional idea that motivated the study (in the form of hypotheses with their respective predictions) is scientifically valid.  Please be more receptive to the comments and opinions of your peers to take full advantage of the academic exchange of the peer review process that is intended to help authors improve their manuscript.  I respect the authors opinion but disagree with their view that the hypothetical-deductive method is an "historic and... out-dated approach to scientific experimentation".  The authors reply to my comment about including hypothesis with: "The hypothesis to test was clear – can C. vicina larvae be reared through to pupation and adult emergence on plant based meal, specifically soya bean meal and canola meal".  But rather than a hypothesis, this phrase looks like the research question that motivated the study.  The possible functional answer to the research question would be the hypothesis.  But this is not clearly stated in the manuscript.

The authors included the following comment from another reviewer in their responses to my comments: "There needs to be more explanation of the BASIS of the selected plant substitutes for meat products. By “basis” I mean a statement of rationale as to how and why the selected plant substitutes were chosen for experimentation. This would involve some kind of statement about the nutritional value of the various components, including some kind of proximate analysis/composition of the products and the diet mixtures. In conjunction with the rationale based on composition/proximate analysis, the paper would profit from a statement of hypothesis as to which combinations could be expected to yield the best responses to the component manipulations".  This comment from another reviewer goes along the lines of my previous comment regarding including a hypothesis.  But neither in the authors' response to this comment, nor in the new (red-colored) text in the revised manuscript, did I find inclusion of the working hypothesis.

Regarding my comments about improving the description of the experimental design and protocol, please do not throw away my opinions just because they are different from those of the other three reviewers.  I am fully aware that authors can always disagree with the view of reviewers, but in such case, the norm is to provide a scientific explanation of why they disagree with one comment or another.  The manuscript still indicates that "An analysis of variance (ANOVA) was performed for normally distributed data...", but as I indicated before, the concept of normality in linear models as is anova, applies for the residuals after fitting the model, not for the raw data.  Also, please be aware that non-parametric Kruskal-Wallis test works with hierarchical data and compares medians, not means.  Therefore, as I stated in my previous review, saying that "the non-parametric Kruskal-Wallis test was employed to compare group means", is misleading.

Another of my comments that was not addressed by the authors without an explanation of why they did so, is the following: "For instance, in Figure 1, the variable "days since oviposition" is presented as a covariate.  But before it is not clearly stated that there was an interest in examining how larval migration and pupation varied across time for each of the diets tested".  This comment remains the same.

In my previous review I suggested to the authors that they include a table with the diet ingredients as a percentage.  But my comment was not taken up by the authors without an explanation from them as to why.  Including the diet formulations is important so that others can replicate the exact same diets used in his study and so that readers have a clear understanding of the exact composition of the diets tested.  I feel that the way that the larval diets are described is not accurate and may confuse readers.  For example, diet T1 is reported as 100% livestock meat meal, but this diet also contains 200 mL of water.  What is not indicated is the mass of the meat meal that as mixed with 200 mL of water and how much diets was prepared.  It is mentioned that five portions of 200 g each were used from each diet.  So, it can be guessed that if 1 kg of each diet was used in the experiments, the authors prepared more than 1 kg of each diet.  But the description of the diet formulations should be clear enough and leave no room for guessing on the part of the readers.  The authors should make a better effort to accurately describe the diet formulations evaluated (T1-T6).  This can be made by including the formulation of the diets in percentages (w/w) and indicate the amount of diet prepared.  For example, if authors prepared 1 kg of each diet, the formulations of diet T1 (with 200 mL of water) would be: 80% (w/w) livestock meat meal and 20% (w/w) water; but if they prepared 1.5 kg of diet, the formulation of diet T1 would be 86.7% (w/w) livestock meat meal and 13.3% (w/w) water.  So again, please be precise and specific in the description of the diets used indicating their formulations (% w/w) so that others can replicate your experimental protocol.  I also suggest that the term "diet" be adopted throughout the manuscript to refer to the treatments employed (i.e., diets T1-T6).

Finally, when replying to reviewers, please indicate the line number of the manuscript in which the change made following reviewers’ comments is found.

Author Response

Comment: Asking authors to include explicit mention of the functional idea that motivated the study (in the form of hypotheses with their respective predictions) is scientifically valid. Please be more receptive to the comments and opinions of your peers to take full advantage of the academic exchange of the peer review process that is intended to help authors improve their manuscript.  I respect the authors opinion but disagree with their view that the hypothetical-deductive method is an "historic and... out-dated approach to scientific experimentation". The authors reply to my comment about including hypothesis with: "The hypothesis to test was clear – can C. vicina larvae be reared through to pupation and adult emergence on plant based meal, specifically soya bean meal and canola meal". But rather than a hypothesis, this phrase looks like the research question that motivated the study. The possible functional answer to the research question would be the hypothesis. But this is not clearly stated in the manuscript.

Response:

A hypothesis has now been stated in the introduction starting on page 3, line 10, viz., “Given the similar nutritional profiles of both soya bean and canola meal with each containing all the essential amino acids, the hypothesis for the basis of this study was that larvae of a necrophagous fly can be reared on these two plant-based meals to pupation and adult eclosion.

The authors included the following comment from another reviewer in their responses to my comments: "There needs to be more explanation of the BASIS of the selected plant substitutes for meat products. By “basis” I mean a statement of rationale as to how and why the selected plant substitutes were chosen for experimentation. This would involve some kind of statement about the nutritional value of the various components, including some kind of proximate analysis/composition of the products and the diet mixtures. In conjunction with the rationale based on composition/proximate analysis, the paper would profit from a statement of hypothesis as to which combinations could be expected to yield the best responses to the component manipulations".  This comment from another reviewer goes along the lines of my previous comment regarding including a hypothesis.  But neither in the authors' response to this comment, nor in the new (red-colored) text in the revised manuscript, did I find inclusion of the working hypothesis.

Response:

In answer to the reviewers comment that “A statement of rationale as to how and why the selected plant substitutes were chosen for the experimentation:, I have already stated on Page 3 line 3 that “the insect rearing industry utilizes soya bean as a major source of protein in the feeds due to their high content and beneficial composition of amino acids” and further on Page 3 line 9 that “Both soya bean and canola meal are one of the few vegetable foods that contain all 9 essential amino acids”.  I have given a complete profile of the amino acid composition of both soya bean and canola meal in Table 2 and stated on Page 4 line 33 that “The amino acids in both soya bean and canola meal protein are close to that of livestock meatmeal, in particular their levels of each essential amino acid. The decision to test plant-based meals with mass rearing of this fly was done to replace typical animal-based sources (eg meatmeal) and various tissues (eg muscle, lung, heart, liver) that have pungent odours when decomposing and require refrigeration.  This has all been mentioned in the introduction and further addressed in the discussion.

In terms of the comment that we should “include some kind of proximate analysis/composition of the products and the diet mixtures, the new Table 1 that I have inserted into the MS (starting page 4 line 26) gives an analysis and/or composition of the products and diet mixtures tested.

Comment: Regarding my comments about improving the description of the experimental design and protocol, please do not throw away my opinions just because they are different from those of the other three reviewers.  I am fully aware that authors can always disagree with the view of reviewers, but in such case, the norm is to provide a scientific explanation of why they disagree with one comment or another.  The manuscript still indicates that "An analysis of variance (ANOVA) was performed for normally distributed data...", but as I indicated before, the concept of normality in linear models as is anova, applies for the residuals after fitting the model, not for the raw data.  Also, please be aware that non-parametric Kruskal-Wallis test works with hierarchical data and compares medians, not means.  Therefore, as I stated in my previous review, saying that "the non-parametric Kruskal-Wallis test was employed to compare group means", is misleading.

Response:

We have changed the statistical analysis description to now read starting after the first 2 sentences (from Page 6 line 2) that: “After fitting the ANOVA model, the normality and homogeneity of variances assumptions were verified. The normal Q-Q probability plot of residuals was used to check that the residuals were normally distributed, while the residuals versus fits plot was employed to assess the homogeneity of variances. Additionally, Bartlett’s test was used to determine the homogeneity of variances across the larval diets. Tukey’s Multiple Comparison test was used to determine which means are significantly different from one another at the 5% significance level. In cases where the variances were not homogenous, the non-parametric Kruskal-Wallis test was employed to compare group means. Subsequently the post-hoc Dunn’s test was conducted for pairwise comparisons to identify significantly different means [53].

Comment: Another of my comments that was not addressed by the authors without an explanation of why they did so, is the following: "For instance, in Figure 1, the variable "days since oviposition" is presented as a covariate.  But before it is not clearly stated that there was an interest in examining how larval migration and pupation varied across time for each of the diets tested".  This comment remains the same.

Response:

In the statistical analysis (at page 5, line 17) it is now stated that Larval migration (i.e., they had left the larval diet and were either under the tray or in the sand), along with newly formed pupae and adult emergence were recorded every day from every larval diet replicate. This allowed us to examine how larval migration and pupation varied across time for each of the diets tested.   

Comment: In my previous review I suggested to the authors that they include a table with the diet ingredients as a percentage.  But my comment was not taken up by the authors without an explanation from them as to why.  Including the diet formulations is important so that others can replicate the exact same diets used in his study and so that readers have a clear understanding of the exact composition of the diets tested.  I feel that the way that the larval diets are described is not accurate and may confuse readers.  For example, diet T1 is reported as 100% livestock meat meal, but this diet also contains 200 mL of water.  What is not indicated is the mass of the meat meal that as mixed with 200 mL of water and how much diets was prepared.  It is mentioned that five portions of 200 g each were used from each diet.  So, it can be guessed that if 1 kg of each diet was used in the experiments, the authors prepared more than 1 kg of each diet.  But the description of the diet formulations should be clear enough and leave no room for guessing on the part of the readers.  The authors should make a better effort to accurately describe the diet formulations evaluated (T1-T6).  This can be made by including the formulation of the diets in percentages (w/w) and indicate the amount of diet prepared.  For example, if authors prepared 1 kg of each diet, the formulations of diet T1 (with 200 mL of water) would be: 80% (w/w) livestock meat meal and 20% (w/w) water; but if they prepared 1.5 kg of diet, the formulation of diet T1 would be 86.7% (w/w) livestock meat meal and 13.3% (w/w) water.  So again, please be precise and specific in the description of the diets used indicating their formulations (% w/w) so that others can replicate your experimental protocol.  I also suggest that the term "diet" be adopted throughout the manuscript to refer to the treatments employed (i.e., diets T1-T6).

Response:

I have inserted an extra Table 1 (below) into the manuscript (starting page 2 line 26) so it is more clear what the diet formulations are and their respective nutritional components (i.e. protein, carbohydrates and fat)

Table 1. Proportion of each ingredient (v/v) in a range of larval diets (T1–T14) tested across three laboratory-based studies using newly hatched larvae of Calliphora vicina. The proportions of protein (P), carbohydrates (C) and fat (F) in each diet are indicated.

% Larval Diet Ingredients (Dry)

Water

Nutritional Profile

Treatment

LMM

WEP

SBM

CM

%P

%CHO

%F

T1

100

200mL

52.0

38.0

10.0

T2

90

10

200mL

51.4

34.5

13.2

T3

100

600mL

47.5

32.5

18.0

T4

10

90

600mL

47.4

29.6

20.4

T5

100

800mL

36.9

33.6

12.0

T6

10

800mL

35.2

30.3

15.0

LMM=Livestock meatmeal; WEP=Whole egg powder; SBM=Soya bean meal; CM=Canola meal;

We have clarified in the Methodology (page 2 line 39) that

“The dry ingredients of each media treatment were first combined (v/v) to make up 1kg of dry media. Water was then added to each dry media blend to make them all up to the same consistency (see Table 1 for amounts added). Five (5) replicate amounts of 200g of the moist media mixture was then placed into rectangular plastic containers (20 cm × 10 cm wide)”.  The amounts of water added to each dry media blend are given in Table 1 (above)

I have used the term “diet” throughout the MS as the reviewer suggests

I have used the term “livestock-derived meatmeal” throughout the MS for consistency

Comment: Finally, when replying to reviewers, please indicate the line number of the manuscript in which the change made following reviewers’ comments is found.

Response:

I have indicated on what page and line number each of these changes has been made

Round 3

Reviewer 4 Report

Comments and Suggestions for Authors

Thank you to the authors for addressing my comments in the revision of their manuscript.  I only have some additional minor comments.

Page 6, line 1: Please remove the statement that “analysis of variance (ANOVA) was performed for normally distributed data”.  As I mentioned in past reviews, the decision to use anova should be based on the nature of the explanatory and response variables.  That is, when the explanatory variables are categorical and the response variables are continuous, anova is used.  It is not necessary to check the normality of the data to decide to use anova.  Normality and homoscedasticity are assumptions that must be checked with the residuals of a model after fitting it to the data.  In addition to the article on the subject by Kéry & Hatfield (2003) that I recommended to the authors in my first review, I also highly recommend the book Crawley, M. J. (2012). The R book. John Wiley & Sons, which I am sure can help authors to improve the accuracy and elegance of their statistical analyses.

In their reply to my comment about the mislead report of Kruskal-Wallis test used to "compare group means", the authors again mention that "the non-parametric Kruskal-Wallis test was employed to compare group means".  But the text was corrected in the manuscript "the non-parametric Kruskal-Wallis test was employed and significant differences were identified using Dunn’s test for pairwise comparisons [53]", so this is OK.

Need to explain how the nutritional profiles of the diet treatments reported in Table 1 were estimated.

In Table 1, please revise the following: (i) T6 is missing 90% of canola meal; (ii) "%CHO" is nuclear, in the figure legend it is indicated "carbohydrates (C)" (not CHO); (iii) indicate in the table that 1 kg of dry ingredients was used for each diet treatment.

Author Response

Comment:

Page 6, line 1: Please remove the statement that “analysis of variance (ANOVA) was performed for normally distributed data”.  As I mentioned in past reviews, the decision to use anova should be based on the nature of the explanatory and response variables.  That is, when the explanatory variables are categorical and the response variables are continuous, anova is used.  It is not necessary to check the normality of the data to decide to use anova.  Normality and homoscedasticity are assumptions that must be checked with the residuals of a model after fitting it to the data.  In addition to the article on the subject by Kéry & Hatfield (2003) that I recommended to the authors in my first review, I also highly recommend the book Crawley, M. J. (2012). The R book. John Wiley & Sons, which I am sure can help authors to improve the accuracy and elegance of their statistical analyses.

Response:

We have amended the text under Statistical Analysis to now read starting Page 6 Line 1:

"All the data collected on larval development to the wandering phase and pupation was analysed using R (version 4.1.1) with the “nlme” and “dplyr” packages [52]. An analysis of variance (ANOVA) was performed using the function “aov” [formula: response variable ~ treatment] to evaluate the effects of each treatment (larval media composition; independent variable) on fly development (% larval wanderers, % pupation; dependent variables). After fitting the ANOVA model, the normality and homogeneity of variances assumptions were verified. The normal Q-Q probability plot of residuals was used to check that the residuals were normally distributed, while the residuals versus fits plot was employed to assess the homogeneity of variances. Additionally, Bartlett’s test was used to determine the homogeneity of variances across treatments. Tukey’s HSD Multiple Comparison test used to determine which means are significantly different from one another at the 5% significance level. In cases where variances were not homogeneous, the Kruskal-Wallis test was employed (non-parametric analysis) and significant differences were identified using Dunn’s test for pairwise comparisons [53]."

Comment:

In their reply to my comment about the mislead report of Kruskal-Wallis test used to "compare group means", the authors again mention that "the non-parametric Kruskal-Wallis test was employed to compare group means".  But the text was corrected in the manuscript "the non-parametric Kruskal-Wallis test was employed and significant differences were identified using Dunn’s test for pairwise comparisons [53]", so this is OK.

Response:

There is no mention of the non-parametric Kruskal-Wallis test being employed to compare group means.

Comment:

Need to explain how the nutritional profiles of the diet treatments reported in Table 1 were estimated.

Response:

I have indicated in the Materials and Methods how I determined the nutritional profiles of the larval diet treatments, starting on Page 5, line 26:

"This information was derived from the nutritional analysis provided by the companies that produce each of the products, e.g., Talloman (Hazelmere WA) for the livestock-derived meatmeal and PBA Foods (Toowoomba, Qld)  foods for the soya bean meal and canola meal.  By knowing the proportion of each ingredient in the larval diets with two ingredients (i.e., T2, T4 and T6), we could determine the %P, CHO and F in the diet blend."

Comment:

In Table 1, please revise the following: (i) T6 is missing 90% of canola meal; (ii) "%CHO" is nuclear, in the figure legend it is indicated "carbohydrates (C)" (not CHO); (iii) indicate in the table that 1 kg of dry ingredients was used for each diet treatment.

Response:

I have changed Table 1 so that all references to carbohydrates is “CHO” in the legend and the table itself are the same.

I have indicated that I have added in “90%” under the column Canola Meal for T6